# Comparative LCA of diesel and grid electricity for agricultural irrigation in Iran: why diesel outperforms in fossil-reliant grids

Majid Namdari *

Department of Plant Production and Genetics, Faculty of Agriculture, University of Zanjan, Zanjan, Iran

* namdari@znu.ac.ir

## Abstract

This study presents a comparative environmental impact assessment of well water extraction systems employed for agricultural irrigation in Iran, focusing on two prevalent energy sources: diesel engines and grid-connected electricity. Utilizing life cycle assessment methodology, the research evaluates key environmental indicators, encompassing abiotic depletion, global warming potential (GWP), human toxicity, aquatic ecotoxicity, acidification, eutrophication, and photochemical oxidation. Data collected from 100 agricultural wells were analyzed to compare the environmental performance of each energy source. The objective is to provide policymakers with data-driven insights for the development of effective mitigation strategies. Environmental analysis revealed that diesel systems emit $1.34\,kg\,CO_2/m^3$, compared to $1.59\,kg\,CO_2/m^3$ in grid systems. While grid-connected systems avoid on-farm emissions, they suffer from high upstream impacts due to Iran's fossil fuel-dominant electricity generation mix (94%). The diesel system showed lower impacts in seven of eleven categories, including GWP, abiotic depletion, and marine ecotoxicity. In contrast, grid systems had slightly lower values in eutrophication ($5.29 \times 10^{-4}$ vs. $1.66 \times 10^{-3}$ kg $PO_4$ eq) and ozone layer depletion. The overall Environmental Composite Index (ECI) was $3.68 \times 10^{-4}$ nPt for diesel and $5.77 \times 10^{-4}$ nPt for electricity, indicating a 56.79% higher burden for the grid-powered system. The findings emphasize the role of local energy mix, pump efficiency, and transmission losses in environmental outcomes, suggesting that improvements in Iran's grid efficiency and a transition to renewable energy are key to reducing impacts.

## 1. Introduction

The escalating demand for water in Iran's agricultural sector, driven by population growth, industrial expansion, and the expansion of irrigated lands, particularly in arid regions with highly variable rainfall patterns, has led to excessive groundwater extraction [1]. Farmers in Iran rely primarily on groundwater for their irrigation needs.

**Data availability statement:** All relevant data are within the paper and its Supporting Information files.

**Funding:** The author(s) received no specific funding for this work.

**Competing interests:** The authors have declared that no competing interests exist.

Approximately 416,000 agricultural deep and semi-deep wells are estimated to exist in Iran, with an annual extraction of greater than 50.2 billion cubic meters of water, which serve as the primary source of irrigation for agricultural production [2]. Among the agricultural wells in Iran, approximately 220,000 are connected to the national electricity grid, while 180,000 rely on diesel fuel for operation. The combustion of fossil fuels in various systems, including diesel engines used for agricultural water pumps, leads to the production of pollutants such as soot and greenhouse gases [3,4]. Intensive groundwater pumping, in addition to depleting reserves, significantly increases energy consumption. This energy typically originates from electricity generation or the combustion of fossil fuels, consequently leading to elevated carbon emissions [5,6]. Therefore, enhancing the efficiency of energy systems emerges as a pivotal strategy for mitigating these emissions and reducing the environmental footprint of agricultural activities [7].

Energy constitutes a fundamental element of contemporary existence, frequently taken for granted by societies benefiting from the conveniences of modern civilization. Concurrently, the magnitude of environmental degradation has surpassed previous estimations. The detrimental effects of substantial consumption of fossil fuel-based energy carriers have exceeded the regenerative capacity of the environment. Climate change has significantly compromised public health, and these ramifications are projected to intensify in the coming decades [8].

The increasing global recognition of the finite nature of petroleum resources and the intensifying crisis of climate change has catalyzed substantial international research and development initiatives focused on the identification and deployment of alternative energy sources for fuel and power generation [9]. The unsustainable exploitation of fossil resources has resulted in elevated greenhouse gas emissions and an accelerated rate of global warming. Consequently, there is a burgeoning demand for sustainable alternatives to diminish reliance on non-renewable energy sources and alleviate adverse environmental consequences [10]. The current utilization of renewable energy sources within irrigation sector remains limited. It should be noted that the development of robust renewable energy platforms is imperative due to the shortage of fossil fuel resources and the detrimental environmental impacts caused by the combustion of petroleum-based fuels [11–13]. At the present time, the Iranian government is offering a number of incentives to those who are converting diesel systems to grid electricity systems [2].

Irrigated agricultural systems exhibit substantial greenhouse gas (GHG) emissions stemming from the energy demand associated with operating irrigation pumps [14]. The choice of suitable primary resources for energy production is a critical determinant of energy efficiency and the overall environmental impact associated with energy systems [15]. The selection of energy sources for powering irrigation equipment exerts a profound influence on the magnitude of these emissions. Consequently, the choice of energy source constitutes a critical factor in determining the overall GHG footprint of irrigation activities. It is crucial to acknowledge that the spatial distribution of these emissions may vary, with emissions potentially occurring both within the farm boundaries (on-farm) and beyond (off-farm). This spatial variability

is contingent upon whether the emissions originate from on-farm energy generation or from the production stages of the utilized energy source [16,17].

According to the most recently available official statistics, 14.2% of Iran's total energy consumption is attributable to the agricultural sector [18]. The overwhelming majority of this consumption is related to water pumping for irrigation operations in the farms [19–21]. Concurrently, data analysis reveals that in numerous Iranian provinces, more than half of the total electricity consumption during the summer season is allocated to pumping water from agricultural wells [1]. This not only exacerbates aquifer depletion but also contributes significantly to increased greenhouse gas emissions, primarily $CO_2$, due to the heightened consumption of fossil fuels for electricity generation [14]. Approximately 10–12 percent of the total emissions worldwide are attributed to the agricultural sector [22]. In Iran, groundwater pumping necessitates the annual consumption of 20.5 billion kWh of electricity and 2 billion liters of fuel, resulting in emissions that constitute 3.6% of the nation's total greenhouse gas emissions [1]. Further studies in Mexico and Pakistan have reported annual carbon emissions of 4.7 and 3.8 million tons, respectively, attributable to groundwater irrigation. These figures represent 3.6% and 2.2% of total carbon emissions in Mexico and Pakistan, respectively [23,24].

Research conducted in China, the world's second largest carbon emitter, has revealed that agricultural irrigation accounted for between 36.72 and 54.16 million tons of carbon dioxide equivalent ($CO_2$-e) emissions in 2010. These findings indicate that energy-related activities in irrigation contribute to 50–70% of total greenhouse gas emissions within the agricultural sector [25]. A study conducted in Egypt assessed carbon emissions associated with diesel and electric on-farm irrigation pumps, utilizing environmental, economic, and social indicators. The findings revealed an average carbon dioxide equivalent emission of 690 tons per cubic meter of water pumped. The analysis demonstrated that electric pumps exhibited superior environmental, economic, and social benefits compared to their diesel counterparts [14]. A study in India, evaluated the techno-economic and environmental aspects of solar photovoltaic (PV), diesel, and electric water pumps for irrigation. The analysis covered performance, cost-effectiveness, and $CO_2$ emissions. The results highlighted the benefits of solar photovoltaic water pumps (SPVWP), such as reduced $CO_2$ emissions, lower operational costs, and the potential for revenue from surplus electricity and carbon trading. The study emphasized the importance of policies to promote SPVWP adoption and identified factors like crop rotation, energy costs, and groundwater availability as key considerations [26]. Ul Hussan et al. [27] conducted a comparative analysis of solar-, electric-, and diesel-powered drip irrigation systems for maize in Pakistan, focusing on economic viability and environmental impact. Their findings revealed that solar power had the lowest $CO_2$ emissions (0.02 tons/ha) and electric power was the most cost-effective (B-C ratio: 1.65). The study emphasized the need for subsidies for solar-powered systems to reduce emissions and address electricity shortages. Koushki et al. [28] compared various energy supply and demand management practices to reduce greenhouse gas emissions from agricultural groundwater pumping. Their results showed that electric pumps, despite higher energy efficiency, produced more GHG emissions than natural gas pumps when using conventional electricity mixes. Improving the operational pump efficiency (OPE) of natural gas pumps resulted in greater GHG savings, and switching to cleaner energy sources like wind and solar offered even higher reductions in emissions. Pradeleix et al. [29] used life cycle assessment to evaluate the environmental impacts of different groundwater pumping systems in semi-arid central Tunisia. Their study found that energy consumption had the greatest environmental impact, particularly on human health, ecosystems, and abiotic depletion. The study highlighted that, in addition to pump efficiency, the energy source significantly influences the environmental performance of pumping systems. Diesel pumps were found to be more harmful than electric pumps when powered by natural gas, but they became more advantageous when electricity was derived from coal, especially with higher pump efficiency. Water depletion was also a critical concern in their findings. Tyson et al. [17] developed a framework for assessing life cycle greenhouse gas emissions and energy consumption in groundwater-irrigated agriculture. Using this framework, they compared GHG emissions and energy use in the Musi catchment in India and the southeast region of South Australia. The study found that the Australian region emitted roughly twice the GHGs per unit of water delivered compared to India, with electricity-powered water pumps contributing over 99% of emissions in

both regions. Interestingly, diesel-powered pumps had the lowest emissions per unit of water supplied. Handa et al. [30] assessed the overall pumping efficiency, greenhouse gas emissions, and costs of irrigation pumping for electric and natural gas-powered pumps in Oklahoma. They found that most pumping plants operated below achievable efficiency levels, with natural gas pumps in the Ogallala aquifer emitting 49% more GHGs than electric pumps in the Rush Springs aquifer. However, natural gas pumps had lower emissions per unit area and head of pumping. The study showed that improving OPE to optimal levels could reduce energy requirements by up to 34% and GHG emissions by up to 52%, highlighting the potential for significant environmental and cost savings through efficiency improvements.

Several studies have examined the use of solar-powered irrigation systems from environmental, economic, and technical perspectives. The results of these studies indicate that solar systems significantly reduce greenhouse gas emissions, fuel costs, and environmental impacts compared to diesel systems [31, 32, 33, 34]. Furthermore, in many regions, these systems have been proposed as a cost-effective and sustainable option for agriculture [35–37].

The control of global warming and the reduction of resource consumption are two of the primary objectives of international environmental policy [38]. According to the findings of the studies, the average temperature of the Earth has been increasing over the past century. The majority of scientists believe that this increase in temperature is a result of increased greenhouse gas emissions. Thus, it is imperative to reduce greenhouse gas emissions in the production process [39]. In consideration of environmental concerns surrounding greenhouse gas emissions and the prospect of global warming, numerous countries that have endorsed the Kyoto Protocol perceive an obligation to curtail these emissions, and have accordingly initiated initiatives in this regard. In order to achieve the aforementioned goals, these countries are attempting to enhance the efficiency of their energy consumption and carbon decomposition while concurrently reducing their reliance on energy-intensive activities [14,40]. The reduction of greenhouse gas emissions within irrigated agricultural systems constitutes a crucial element in enhancing both mitigation and adaptation capacities to climate change [14].

The environmental impact of combustion systems, such as those used in diesel engines, extends beyond greenhouse gas emissions [41]. Life cycle assessment (LCA) is one of the environmental impact assessment methods that examines the environmental effects of a product or process by identifying, measuring, and evaluating all consumed resources, emissions, outputs, and waste [22]. LCA is a technique for evaluating the potential environmental effects of a product's production. It is accomplished by compiling an inventory of the product's environmental interactions and evaluating the corresponding potential environmental impacts during the product's life cycle [42]. LCA has gained a prominent position within the agricultural sector due to its capacity to assess the environmental consequences of products throughout their entire life cycle, from production to consumption (cradle-to-grave study). Additionally, it enables the differentiation of input consumption and environmental impact between diverse systems [43].

However, due to Iran's abundant access to fossil fuel resources, the country has not demonstrated sufficient commitment to the widespread use of renewable energy sources such as solar systems. Furthermore, the lack of sunlight during nighttime hours presents a major challenge for solar systems, as irrigation operations are often more efficient during these periods. Consequently, farmers often prefer to use grid-connected or diesel systems, which allow for continuous operation. As a result, diesel and grid-connected systems are currently the most common methods for extracting water from wells in Iran. Moreover, in recent years, the government has provided numerous incentives and subsidies to replace diesel systems with grid-connected systems. Given these circumstances, the primary objective of this study is to compare the environmental impacts of diesel and grid-connected systems for groundwater extraction from agricultural wells using life cycle assessment. This research aims to provide insights to decision-makers and policymakers by comparing the environmental impacts of these two methods, enabling them to make informed decisions to mitigate negative environmental effects.

## 2. Materials and methods

This research employed the life cycle assessment methodology to evaluate the environmental burdens associated with the extraction of one cubic meter of water from agricultural wells, comparing two energy sources: electricity from the

national grid and diesel fuel. This investigation adhered to the ISO standards 14040 [44] and 14044 [45], encompassing four sequential stages: goal and scope definition, life cycle inventory (LCI) analysis, life cycle impact assessment (LCIA), and interpretation of results. Detailed explanations of each phase are provided in the following sections.

### 2.1. Goal and scope definition: site description, evaluated scenarios, functional unit, and system boundaries

In this study, a sample size of 100 wells was selected to achieve a representative and statistically robust dataset while considering practical constraints. The water pumping systems of these wells were converted from diesel to electric (national grid electricity) between 2017 and 2018; therefore, data were collected over the period from 2016 to 2020. The energy consumption data, including diesel fuel consumption and electricity consumption, were recorded for both configurations. The wells were distributed across Tehran, Hamadan, Kurdistan, and Fars to ensure regional diversity and representativeness. These provinces cover a wide range of climatic conditions, agricultural practices, and energy infrastructure, providing a comprehensive analysis of Iran's agricultural landscape. The scenarios analyzed included a diesel-powered water pumping system (Scenario I) and a grid electricity-powered water pumping system (Scenario II). Figs 1 and 2 provide visual representations of these scenarios. Due to the limited number of wells utilizing renewable energy sources,

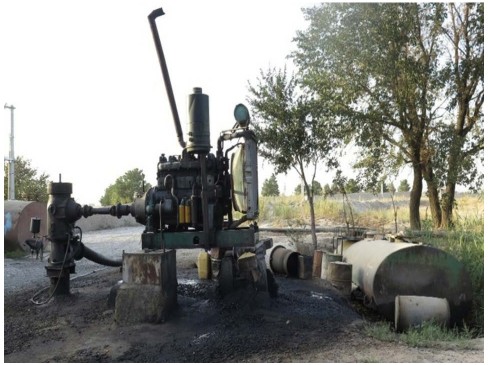

**Fig 1. Diesel-powered water extraction (Scenario I).**

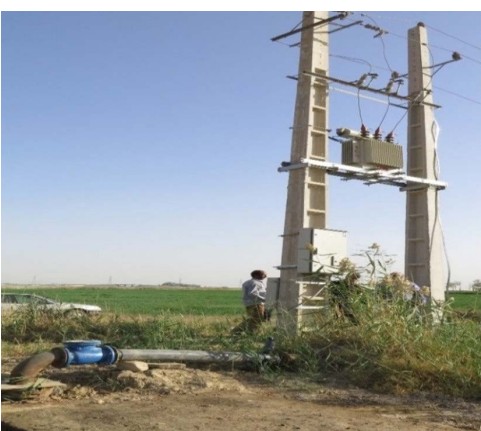

**Fig 2. Grid-powered water extraction (Scenario II).**

 

such as solar or wind energy, and the preference of farmers for grid electricity or diesel systems, the study did not investigate solar or wind energy systems, although further studies on these alternatives are recommended.

In the diesel pumping system, the diesel fuel produced at refineries is transferred to storage tank facilities in oil depots across each province via fuel pipelines or fuel delivery trucks. The diesel is then sent from the central depots to agricultural fuel distribution centers through fuel tankers. Farmers subsequently purchase this fuel from the distribution centers, which is transported to their farms by delivery tankers. The fuel is stored in tanks on the farms and is gradually consumed. This system includes a diesel engine, which provides the necessary power for the water pump. In Iran, water is generally extracted from wells using a deep-well turbine system (Fig 3).

In the electric pumping system, electricity is generated in power plants. Power plants in Iran are generally thermal, powered by fossil fuels including gas, diesel, mazut, or coal. The electricity is transmitted through transmission towers (either overhead or underground lines) to regional distribution substations. In local stations, the voltage of the electricity is reduced to enter urban and rural distribution networks. Subsequently, electricity is transferred from the main lines to rural converters and agricultural farms. Then, electricity enters the agricultural pump house through low-voltage cables and with the assistance of utility poles, which are generally made of concrete. Inside the pump house, which is equipped with a transformer, the electricity is transferred to a water pump, typically a submersible pump, which facilitates water extraction (Fig 4).

The system boundaries for both scenarios were defined using a cradle-to-use approach. For diesel-powered systems, this included diesel fuel production, distribution, storage, on-farm combustion, and the manufacturing of diesel engines and pumps, while excluding maintenance activities. For grid-connected systems, the boundaries encompassed electricity generation, transmission and distribution losses, electricity consumption, and the manufacturing of electrical equipment (e.g., pumps, transformers), while excluding maintenance activities. Specifically, the system boundary extended until the energy input enabled the extraction of one cubic meter of water (functional unit) from the designated wells. The type of water distribution network was not among the objectives of this study.

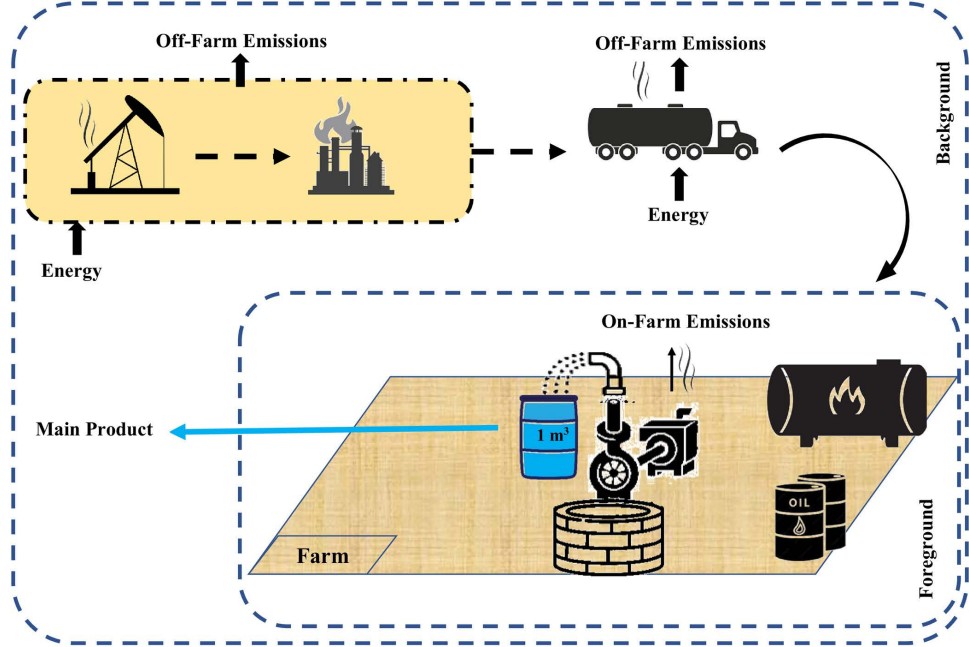

**Fig 3. System boundaries of diesel-powered water extraction.**

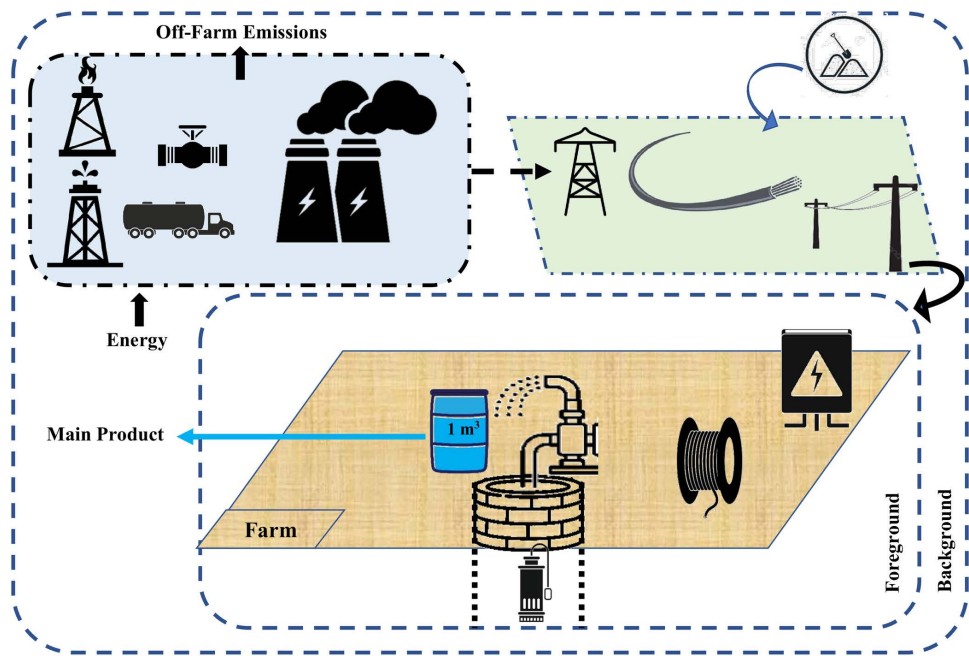

**Fig 4. System boundaries of electric-powered water extraction.**

## 2.2. Life cycle inventory analysis

Data collection for LCI constitutes a significant source of uncertainty and requires substantial effort within the framework of LCA studies. This critical stage involves meticulously quantifying and comprehensively cataloging all inputs and outputs that interact with the system under investigation [21].

Scenario I includes the stages of diesel fuel production, distribution, storage, and consumption for water extraction from agricultural wells. The fuel consumption rate was determined precisely based on interviews with farmers and fuel purchase records. The fuel production stage at the refinery was modeled using data from Ecoinvent 3, while the distribution, storage, and consumption stages were inventoried for each well, considering its specific conditions, based on interviews with farmers and official data obtained from the Agricultural Organization and the National Oil Products Distribution Company. In this process, parameters such as distance, number and type of fuel transport pipelines, fuel transfer and distribution methods for each well, fuel consumption during transportation operations, storage tank capacity and type, water pump characteristics and deep-well turbine system specifications, and the amount of engine oil consumed by the diesel engine were identified and included in the life cycle inventory. Table 2 presents this inventory, where the reported values represent the average amounts for each input. During the fuel combustion phase on the farm, pollutant emissions were recorded as outputs in the LCI, and these emissions were quantified based on the emission factors provided in IPCC reports [46].

Scenario II includes the stages of electricity generation, transmission, distribution, and consumption in electric motors driving water pumps in agricultural wells. Iran's electricity grid composition, which is 94% reliant on fossil fuels, was modeled using official statistics from the International Energy Agency (IEA) for the year 2025. The grid mix includes natural gas (80.21%), oil (13.54%), coal (0.19%), and renewables/nuclear (6.06%). The environmental impacts of electricity generation were calculated based on the emissions and resource consumption associated with each energy source. The life cycle inventory for transmission, distribution, and consumption was established for each well based on its geographical location

and distance from electricity distribution centers, as well as interviews with farmers and electricity infrastructure contractors. Implementation maps of rural electrification projects, which provide detailed information on the consumption of each input, were utilized in the inventory process. The LCI in this scenario includes inputs such as power poles, cables, transformers, sand, gravel, and cement (for installing power poles). Additionally, near the well site, the water pump and HDPE pipes were incorporated into the inventory. Unlike Scenario I, on-farm emissions are generally absent in Scenario II, while most emissions occur off-farm. Electricity consumption was recorded based on meter readings installed at the wells.

To mitigate potential biases in data collection, diesel consumption was verified through fuel purchase records and interviews with farmers, while electricity consumption was measured using meter readings. Data from multiple sources were cross-verified to ensure accuracy and reliability. All data collection methods and limitations are documented to ensure transparency and reproducibility.

The volume of water consumed by each well was determined with high precision through the use of a meter installed on the wells themselves, allowing for a precise measurement of water usage. The recorded data on water, diesel fuel, and electricity were related to the amount of consumption over the course of one agricultural year.

## 2.3. Life cycle impact assessment

LCIA serves as a crucial step in evaluating environmental burdens identified during the LCI phase. LCIA methodologies facilitate the assessment and interpretation of the potential environmental impacts associated with a given product system. This critical phase involves a comprehensive process encompassing the categorization, characterization, and interpretation of environmental flows, enabling a quantitative understanding of the magnitude and significance of potential impacts [45].

A variety of methodologies are available for evaluating environmental impacts. This research utilized the CML (Centrum voor Milieukunde Leiden) methodology, developed by the Institute of Environmental Sciences at Leiden University. The CML methodology was selected for this study due to its objectivity and international recognition in environmental impact assessment. Unlike other frameworks, which incorporate subjective weighting factors, the CML methodology relies on scientific and objective criteria, making it suitable for comparative studies [47,48].

In this study, the CML methodology was employed to evaluate a comprehensive set of environmental impact categories, which were selected based on their relevance to the agricultural water pumping systems in Iran. These categories include Abiotic Depletion (ADP), Abiotic Depletion (Fossil Fuels) (ADF), Global Warming Potential (GWP), Ozone Layer Depletion Potential (ODP), Human Toxicity (HT), Freshwater Aquatic Ecotoxicity (FWAE), Marine Aquatic Ecotoxicity (MAE), Terrestrial Ecotoxicity (TE), Photochemical Oxidation (PO), Acidification Potential (AP), and Eutrophication Potential (EP). Each of these indicators provides insights into different aspects of the environmental burden associated with water extraction, enabling a holistic assessment of the diesel and grid electricity-powered systems.

The World 2000 method was utilized for normalization and weighting procedures in this study. A composite environmental index was calculated for each well by aggregating the normalized and weighted scores of the individual impact categories. This study utilized SimaPro software (version 9.5.0) to assess a range of environmental impact categories. The SimaPro software was chosen for its comprehensive database and flexibility in handling complex systems, ensuring robust modeling of diesel-powered and grid-connected irrigation systems. The software's output data was subsequently transferred to Microsoft Excel for improved data visualization and analysis. Table 1 presents the specific impact categories examined within this LCA model, including their respective nomenclature and units of measurement.

## 2.4. Interpretation

The final stage of the life cycle assessment methodology, known as life cycle interpretation, represents a structured approach for identifying, quantifying, verifying, and evaluating the information derived from the LCI and LCIA phases. The overarching goal of this stage is to effectively communicate the findings of the LCA [49].

**Table 1. The environmental impact categories and their measurement units.**

| Impact categories | Nomenclature | Measurement units |
|---|---|---|
| Global Warming Potential | GWP | kg $CO_2$ eq |
| Acidification Potential | AP | kg $SO_2$ eq |
| Eutrophication Potential | EP | kg $PO_4^{3-}$ eq |
| Ozone Layer Depletion Potential | ODP | kg CFC-11 eq |
| Abiotic Depletion Potential | ADP | kg Sb eq |
| Abiotic Depletion Potential – Fossil | ADF | MJ |
| Human Toxicity Potential | HT | kg 1,4-DB eq |
| Freshwater Aquatic Ecotoxicity Potential | FWAE | kg 1,4-DB eq |
| Marine Aquatic Ecotoxicity Potential | MAE | kg 1,4-DB eq |
| Terrestrial Ecotoxicity Potential | TE | kg 1,4-DB eq |
| Photochemical Oxidant Formation Potential | PO | kg $C_2H_4$ eq |

## 3. Results and discussion

This study's findings are structured into three sections. Section one details the environmental indicators associated with groundwater extraction powered by diesel engines, designated as Scenario I. Section two quantifies and presents the environmental impact of groundwater extraction utilizing grid electricity, designated as Scenario II. Finally, a comparative analysis of both scenarios is conducted to determine the optimal system.

Life cycle inventory data for both scenarios are presented in Table 2, preceding the presentation and analysis of the LCAI results. The results revealed that the average annual extracted water volume was consistent across both scenarios, registering 149,580 m³ per agricultural year. This extraction was achieved in Scenario I through the utilization of 71,586 liters of diesel fuel, while Scenario II employed 286,729 kWh of electricity. A detailed breakdown of other input resource consumption for each scenario is presented in Table 2. It is crucial to emphasize that each agricultural well operates under a government-mandated annual water extraction quota, and any exceedance of this allocation is strictly prohibited.

In Scenario I, the diesel-powered extraction process not only consumed the aforementioned inputs but also generated emissions due to fuel combustion. Specifically, each well's annual water extraction resulted in the emission of approximately 159,351 kg of carbon dioxide ($CO_2$) and 395 kg of carbon monoxide (CO). Additional on-farm emissions are detailed in Table 2. Conversely, the electrical energy utilized for water extraction in Scenario II does not generate direct on-farm emissions; these emissions are typically associated with off-farm power generation processes.

### 3.1. Environmental impact assessment of water extraction using a diesel system (Scenario I)

Table 3 presents the life cycle impact assessment results for the diesel-powered water extraction system. The values reported represent the impacts associated with the extraction of one cubic meter of water (functional unit, FU) from the wells.

Table 3 presents the aggregated impact assessment results for each category, reflecting the cumulative contributions of all inventory inputs. Each input parameter exerts a distinct and quantifiable influence on the defined impact categories. Fig 5 illustrates the proportional contribution of each input to these categories, expressed as percentages. As shown in the figure, diesel fuel demonstrates a dominant influence across the majority of impact categories. Specifically, it accounts for 32.8% of abiotic depletion, 97.8% of abiotic depletion (fossil fuels), 17.7% of global warming potential, 98.2% of ozone layer depletion potential, 19.1% of human toxicity, 30.4% of freshwater aquatic ecotoxicity, 61.2% of marine aquatic ecotoxicity, 40.9% of terrestrial ecotoxicity, 57.9% of photochemical oxidation, 21.4% of acidification, and 9.75% of eutrophication potential. Critically, these percentages reflect the environmental burdens associated with diesel fuel production

**Table 2. Inventory list of Scenarios I and II.**

| Scenario I | average value | Scenario II | average value |
|---|---|---|---|
| **Inputs (unit)** | | **Inputs (unit)** | |
| water ($m^3$) | 149580 | water ($m^3$) | 149580 |
| Diesel Fuel for pumping (L) | 71856 | Electricity (kWh) | 286729 |
| Engine Oil (L) | 50 | Utility pole# (p) | 0.34 |
| Fuel storage tanks (galvanized steel)* (kg) | 62.5 | Crushed gravel (kg) | 34.29 |
| Pipeline for petroleum** (m) | 0.54 | Sand (kg) | 34.2857 |
| Water pump*** (kg) | 63 | Cement (kg) | 17.14 |
| Deep-well turbine* (kg) | 204 | HDPE pipe (kg) | 54.8 |
| Diesel (for transportation) (L) | 251.16 | Power Transformer## (kg) | 32.4 |
| petrol (for transportation) (L) | 126.28 | Electrical cable (m) | 30.4 |
| Engine oil (for transportation) (L) | 0.21 | | |
| Transport (bulk fuel tankers) (tkm) | 3708.068 | | |
| Transport (delivery tankers) (tkm) | 1506.34 | | |
| **Outputs (emissions****) (unit)** | | **Outputs (emissions) (unit)** | |
| Carbon dioxide (kg) | 159351.31 | | |
| Carbon monoxide (g) | 395237.24 | | |
| Non-methane volatile organic compounds (g) | 97728.41 | | |
| Nitrogen oxides (g) | 1679799.30 | | |
| Particulates (g) | 46997.72 | | |
| Dinitrogen monoxide (g) | 2582.71 | | |
| Ammonia (g) | 723.41 | | |
| **Product (Unit)** | | **Product (Unit)** | |
| Extracted water ($m^3$) | 149580 | Extracted water ($m^3$) | 149580 |

\* Useful Life = 20 y
\*\* Useful Life = 30 y
\*\*\* Useful Life = 15 y
\*\*\*\* Total on-farm and transport emissions

\# Useful Life = 35 y
\#\# Useful Life = 25 y

**Table 3. Impact categories per functional unit in the diesel-powered system.**

| Impact categories | unit | Value |
|---|---|---|
| Abiotic Depletion | kg Sb eq | $3.76 \times 10^{-7}$ |
| Abiotic depletion (fossil fuels) | MJ | 21.3 |
| Global Warming Potential | kg $CO_2$ eq | 1.34 |
| Ozone Layer Depletion Potential | kgCFC-11 eq | $1.01 \times 10^{-7}$ |
| Human toxicity | kg 1,4-DB eq | 0.154 |
| Fresh water aquatic ecotoxicity | kg 1,4-DB eq | 0.04 |
| Marine aquatic ecotoxicity | kg 1,4-DB eq | 97.2 |
| Terrestrial ecotoxicity | kg 1,4-DB eq | $1.79 \times 10^{-4}$ |
| Photochemical oxidation | kg $C_2H_4$ eq | $1.94 \times 10^{-4}$ |
| Acidification Potential | kg $SO_2$ eq | $7.34 \times 10^{-3}$ |
| Eutrophication Potential | kg $PO_4^{3-}$ eq | $1.66 \times 10^{-3}$ |

(background system), rather than its direct consumption (foreground system). Thus, the impacts quantified here arise from the extraction, purification, and refining processes of diesel fuel at the refinery, and are distinct from those resulting from its on-farm use.

On-farm diesel fuel consumption, resulting from the operation of various diesel engines, generates emissions categorized as foreground emissions. These are visually represented by the blue areas in Fig 5. Analysis of the figure reveals that on-farm emissions contribute significantly to several environmental impact categories. Specifically, they account for 88.1% of the eutrophication potential, 80% of the global warming potential, 76.6% of acidification, 36.7% of photochemical oxidation, and 8.74% of human toxicity.

The production of the deep-well turbine pump constitutes another significant input within the assessed categories. This input accounts for a substantial proportion of the observed impacts across several environmental categories, specifically abiotic depletion (55.4%), human toxicity (63.3%), freshwater aquatic ecotoxicity (48.7%), marine aquatic ecotoxicity (22.1%), and terrestrial ecotoxicity (53.6%).

Furthermore, the production of diesel engines contributes 14.0% to freshwater aquatic ecotoxicity and 10.8% to marine aquatic ecotoxicity. As illustrated in Fig 5, the influence of other inputs on the assessed impact categories is comparatively negligible.

To enable a comparative analysis of impact categories and the subsequent derivation of a composite index, normalization and weighting of these categories are required. Table 4 presents the normalized and weighted values for each impact category.

Table 4 presents the comparative analysis of eleven environmental impact categories associated with the extraction of one cubic meter of water. The highest impact score is observed for marine aquatic ecotoxicity ($1.65 \times 10^{-4}$ nPt), significantly exceeding the values observed in other categories. Human toxicity represents the second highest impact ($5.98 \times 10^{-5}$ nPt). Further contributing factors of note include abiotic depletion (fossil fuels) ($6.15 \times 10^{-4}$ nPt), global warming potential ($3.20 \times 10^{-5}$ nPt), and acidification ($3.07 \times 10^{-5}$ nPt). Conversely, terrestrial ecotoxicity demonstrates the lowest impact score among the assessed categories ($5.40 \times 10^{-8}$ nPt).

A prerequisite for the development of effective environmental impact mitigation strategies is the quantification of individual input contributions to diverse impact categories and their cumulative effect. Fig 6 presents a single-score analysis, illustrating the normalized and weighted input values to provide a comprehensive assessment of their aggregated environmental burden.

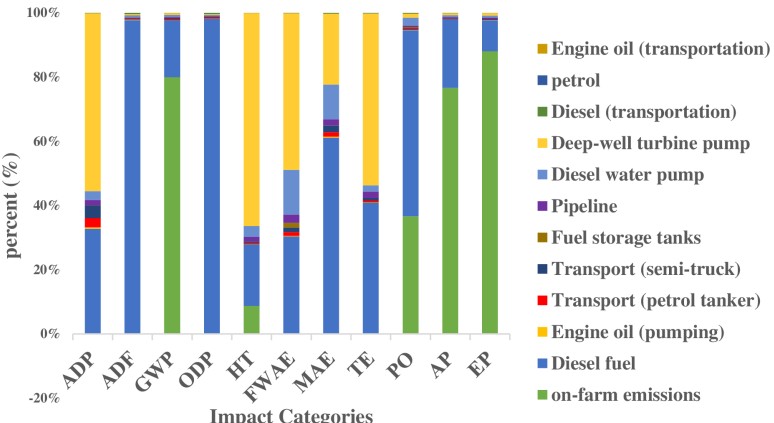

**Fig 5. Impact categories of the diesel-powered water extraction system (Scenario I).**

**Table 4. Normalized and weighted impact category values per cubic meter of water extracted (FU) in the diesel-powered system.**

| Impact categories | normalized value | weighted value (nPt) |
|---|---|---|
| Abiotic Depletion | $1.80 \times 10^{-15}$ | $1.80 \times 10^{-6}$ |
| Abiotic depletion (fossil fuels) | $5.61 \times 10^{-14}$ | $5.61 \times 10^{-5}$ |
| Global Warming Potential | $3.20 \times 10^{-14}$ | $3.20 \times 10^{-5}$ |
| Ozone Layer Depletion Potential | $4.43 \times 10^{-14}$ | $4.43 \times 10^{-7}$ |
| Human toxicity | $5.98 \times 10^{-14}$ | $5.98 \times 10^{-5}$ |
| Fresh water aquatic ecotoxicity | $1.69 \times 10^{-14}$ | $5.58 \times 10^{-6}$ |
| Marine aquatic ecotoxicity | $5.01 \times 10^{-13}$ | $1.65 \times 10^{-4}$ |
| Terrestrial ecotoxicity | $1.64 \times 10^{-16}$ | $5.40 \times 10^{-8}$ |
| Photochemical oxidation | $5.29 \times 10^{-15}$ | $5.29 \times 10^{-6}$ |
| Acidification Potential | $3.07 \times 10^{-14}$ | $3.07 \times 10^{-5}$ |
| Eutrophication Potential | $1.05 \times 10^{-14}$ | $1.05 \times 10^{-5}$ |
| Environmental Composite Index | | $3.68 \times 10^{-4}$ |

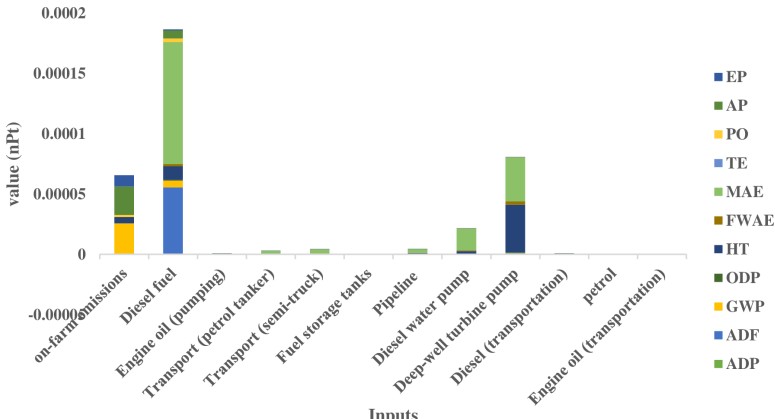

**Fig 6. Single score analysis of environmental impacts for the diesel-based water extraction system.** The figure illustrates the aggregated environmental burden (in nPt) per cubic meter of water extracted. Diesel fuel production and on-farm emissions are the primary contributors to the overall impact.

Fig 6 illustrates the relative environmental impacts of various inputs associated with the operation of agricultural wells. Diesel fuel production, encompassing extraction, refining, and purification processes, exhibits the most substantial environmental impact. Furthermore, the fuel consumption of these wells surpasses established standards, a consequence of employing outdated diesel engines characterized by lower efficiency compared to contemporary models. This operational inefficiency contributes to elevated on-farm emissions, given the direct correlation between diesel consumption and pollutant release. As depicted in Fig 6, on-farm emissions, primarily attributable to diesel combustion, constitute the third most influential factor across the evaluated environmental impact categories. Analysis of well equipment reveals the deep-well turbine pump and the diesel pumping engine as the second and fourth most significant contributors, respectively. Conversely, the environmental impact of other input factors is comparatively negligible.

Following the normalization and weighting of impact category indicators, a composite environmental index was derived for the analyzed scenario. The resulting index value is presented in Table 4. Specifically, the composite environmental

index for the diesel pumping system, expressed per cubic meter of water delivered, was calculated to be $3.68 \times 10^{-4}$. This index facilitates a comparative assessment of the studied system against alternative systems.

## 3.2. Environmental impact assessment of water extraction using grid electricity (Scenario II)

Parallel to the evaluation procedures implemented for the diesel system, assessments were also conducted for the grid-connected water extraction system. Table 5 presents the impact categories associated with Scenario II. As shown in Table 5, the global warming potential for this system is $1.59$ kg $CO_2$-eq.

Fig 7 presents the contribution of each input to the examined impact categories. As illustrated in the figure, electricity production at the power plant constitutes the most significant contributor across all impact categories. Specifically, its contribution amounts to 59% for abiotic depletion, 99.8% for abiotic depletion (fossil fuels), 99.7% for global warming potential, and 99.3% for ozone layer depletion potential. Additionally, electricity accounts for 87.2% of human toxicity, 81.7% of freshwater aquatic ecotoxicity, 86.3% of marine aquatic ecotoxicity, 99.8% of terrestrial ecotoxicity, 99.1% of photochemical oxidation, 98.9% of acidification potential, and 92% of eutrophication potential.

While other inputs have a relatively minor impact on these categories, notable contributions were observed for certain components. Specifically, electric cables and power transformers contribute 33.3% and 7.42%, respectively, to abiotic depletion. Furthermore, electric cables were found to account for 13.2% of freshwater aquatic ecotoxicity, 10.7% of marine aquatic ecotoxicity, 6.13% of eutrophication potential, and 5.58% of human toxicity.

**Table 5. Impact categories in the grid-powered system per cubic meter of water (FU).**

| Impact categories | unit | Value |
|---|---|---|
| Abiotic Depletion | kg Sb eq | $6.23 \times 10^{-7}$ |
| Abiotic depletion (fossil fuels) | MJ | 24.20 |
| Global Warming Potential | kg$CO_2$ eq | 1.59 |
| Ozone Layer Depletion Potential | kgCFC-11 eq | $4.90 \times 10^{-8}$ |
| Human toxicity | kg 1,4-DB eq | 0.137 |
| Fresh water aquatic ecotoxicity | kg 1,4-DB eq | 0.0624 |
| Marine aquatic ecotoxicity | kg 1,4-DB eq | 216 |
| Terrestrial ecotoxicity | kg 1,4-DB eq | $1.40 \times 10^{-3}$ |
| Photochemical oxidation | kg $C_2H_4$ eq | $3.27 \times 10^{-4}$ |
| Acidification Potential | kg $SO_2$ eq | $7.24 \times 10^{-3}$ |
| Eutrophication Potential | kg $PO_4$ eq | $5.29 \times 10^{-4}$ |

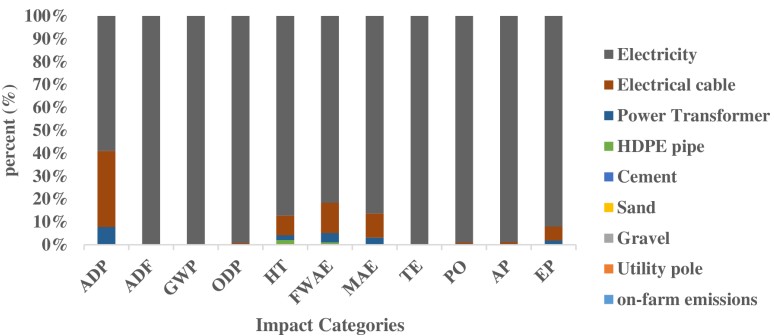

**Fig 7. Impact Categories in the Grid-Powered System (Scenario II).**

In Scenario II, electricity generation constituted the most significant contributor to the overall impact. This finding underscores the critical need for improvements in power generation efficiency to mitigate environmental impact.

Table 6 presents the normalized and weighted results for the impact categories evaluated in Scenario II. The Environmental Composite Index (ECI) for the water extraction system powered by the electricity grid is $5.77 \times 10^{-4}$, as shown in Table 6. Analysis of these results (Table 6) reveals that marine aquatic ecotoxicity, abiotic depletion (fossil fuels), global warming potential, and acidification represent the impact categories with the highest values. Conversely, human toxicity exhibits the lowest impact.

Fig 8 illustrates the contribution of each input to the overall environmental impact, as represented by the Single Score analysis. According to this diagram, electricity generation at the power plant emerges as the dominant contributor across all impact categories, accounting for the highest environmental burden. In contrast, the contributions of the remaining inputs are comparatively minor, with only specific components, such as electric cables and power transformers, exhibiting notable but relatively smaller impacts. Although the contributions of inputs such as utility poles or HDPE pipes are very small, research has shown that advanced materials, such as carbon nanotube-reinforced cementitious composites, can enhance structural durability and limit $CO_2$ diffusion, contributing to more sustainable infrastructure solutions [50].

**Table 6. Normalized and weighted impact category values per cubic meter of water extracted in the grid-powered system.**

| Impact categories | Normalized value | weighted value (nPt) |
|---|---|---|
| Abiotic Depletion | $2.98 \times 10^{-15}$ | $2.98 \times 10^{-6}$ |
| Abiotic depletion (fossil fuels) | $6.35 \times 10^{-14}$ | $6.35 \times 10^{-5}$ |
| Global Warming Potential | $3.81 \times 10^{-14}$ | $3.81 \times 10^{-5}$ |
| Ozone Layer Depletion Potential | $2.16 \times 10^{-16}$ | $2.16 \times 10^{-5}$ |
| Human toxicity | $5.30 \times 10^{-14}$ | $5.30 \times 10^{-14}$ |
| Fresh water aquatic ecotoxicity | $2.64 \times 10^{-14}$ | $8.71 \times 10^{-6}$ |
| Marine aquatic ecotoxicity | $1.11 \times 10^{-12}$ | $3.68 \times 10^{-4}$ |
| Terrestrial ecotoxicity | $1.28 \times 10^{-15}$ | $4.23 \times 10^{-7}$ |
| Photochemical oxidation | $8.89 \times 10^{-15}$ | $8.89 \times 10^{-6}$ |
| Acidification Potential | $3.03 \times 10^{-14}$ | $3.03 \times 10^{-5}$ |
| Eutrophication Potential | $3.34 \times 10^{-15}$ | $3.34 \times 10^{-6}$ |
| Environmental Composite Index | | $5.77 \times 10^{-4}$ |

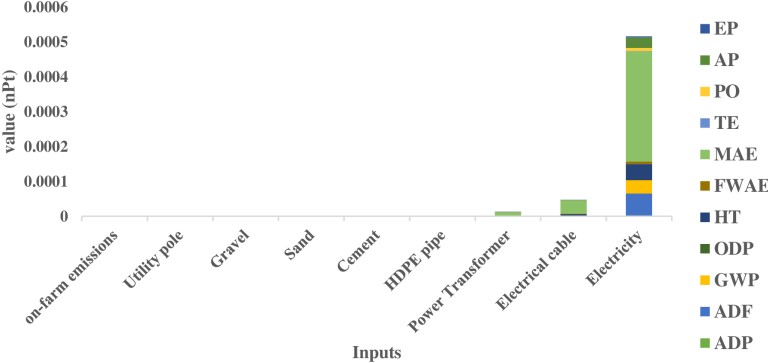

**Fig 8. Single score analysis of environmental impacts for the grid-connected water extraction system.** The figure illustrates the aggregated environmental burden (in nPt) per cubic meter of water extracted. Electricity generation at power plants is the dominant contributor to the overall impact.

## 3.3. Comparative analysis of environmental impacts: diesel vs. electricity-powered systems

Following the assessment of the environmental impacts of both systems, a comparative analysis was performed to evaluate their relative sustainability. This comparative evaluation facilitated the identification of the system exhibiting the lower overall environmental burden, thereby informing the selection of the more environmentally preferable option. Table 7 presents a comparative analysis of the environmental impacts of the two systems under investigation across various impact categories. The table reveals a disparity in abiotic depletion, with the diesel-based system exhibiting a value of $3.76 \times 10^{-7}$ kg Sb eq, while the grid electricity system demonstrates a higher value of $6.23 \times 10^{-7}$ kg Sb eq. This difference in abiotic depletion underscores the distinct environmental burdens associated with each system.

A key observation from Table 7 is the global warming potential associated with each system. The diesel-based system exhibits a GWP of 1.34 kg $CO_2$-eq, while the grid electricity system's GWP is slightly higher at 1.59 kg $CO_2$-eq. This difference underscores the variation in greenhouse gas emissions between the two water extraction methods, highlighting the environmental implications of each energy source. These findings align with those of Tyson et al. [17], who reported that diesel-powered water pump irrigation systems produced the lowest greenhouse gas emissions per unit volume of water supplied compared to grid electricity and diesel generator-powered submersible pumps, further reinforcing the conclusion that diesel-powered systems demonstrate lower overall emissions in groundwater-based irrigation. Similarly, Koushki et al. [28] investigated the comparative environmental impacts of electric and natural gas-powered water pumps, finding that in regions with electricity grids reliant on high-emission energy sources (e.g., coal and natural gas), natural gas-powered pumps exhibit lower greenhouse gas emission profiles than their electric counterparts. Ul Hussan et al. [27], in their examination of diesel- and electric-powered irrigation systems, also reported higher carbon dioxide emissions in the electric system.

Conversely, several studies have reported divergent results. El-Gafy and El-Bably [14], in their study on the environmental, economic, and social impacts of diesel and electric on-farm irrigation pumps in Egypt, concluded that electric pumps are more advantageous across all three aspects. Terang and Baruah [26] reported that, in Indian agricultural crop cultivation, electric-powered pumping systems demonstrated lower carbon dioxide emissions than their diesel counterparts.

These observed inconsistencies may be attributed to variations in energy source, a factor explored in the subsequent section. Koushki et al. [28] demonstrated the influence of energy source on environmental emissions associated with water extraction for irrigation. Their research indicated that both the primary energy source (e.g., fossil fuels or electricity)

**Table 7. Comparative results of impact categories in diesel-powered and grid-powered systems.**

| Impact categories | Unit per FU | Scenario I | Scenario II |
|---|---|---|---|
| Abiotic Depletion | kg Sb eq | $3.76 \times 10^{-7}$ | $6.23 \times 10^{-7}$ |
| Abiotic depletion (fossil fuels) | MJ | 21.3 | 24.20 |
| Global Warming Potential | kg$CO_2$ eq | 1.34 | 1.59 |
| Ozone Layer Depletion Potential | kgCFC-11 eq | $1.01 \times 10^{-7}$ | $4.90 \times 10^{-8}$ |
| Human toxicity | kg 1,4-DB eq | 0.154 | 0.137 |
| Fresh water aquatic ecotoxicity | kg 1,4-DB eq | 0.04 | 0.0624 |
| Marine aquatic ecotoxicity | kg 1,4-DB eq | 97.2 | 216 |
| Terrestrial ecotoxicity | kg 1,4-DB eq | $1.79 \times 10^{-4}$ | $1.40 \times 10^{-3}$ |
| Photochemical oxidation | kg $C_2H_4$ eq | $1.94 \times 10^{-4}$ | $3.27 \times 10^{-4}$ |
| Acidification Potential | kg $SO_2$ eq | $7.34 \times 10^{-3}$ | $7.24 \times 10^{-3}$ |
| Eutrophication Potential | kg $PO_4$ eq | $1.66 \times 10^{-3}$ | $5.29 \times 10^{-4}$ |

and, in the case of electricity utilization, the composition of the electricity grid (i.e., the proportion of coal, natural gas, oil, wind, and solar generation) significantly affect emissions profiles in water pumping systems.

The results presented herein are context-specific, pertaining primarily to Iran and nations with analogous grid electricity infrastructures characterized by a reliance on fossil fuel-based thermal power generation. International variations in power plant configurations, influenced by primary energy source and prevailing operational conditions, are significant. Consequently, it is anticipated that the reported findings will diverge in contexts where renewable energy sources constitute a substantial proportion of grid electricity supply.

A comparative analysis of impact categories between the two systems is presented in Fig 9. This figure demonstrates that abiotic depletion, abiotic depletion (fossil fuels), global warming potential, freshwater aquatic ecotoxicity, marine aquatic ecotoxicity, terrestrial ecotoxicity, and photochemical oxidation exhibit higher values within the grid electricity system compared to the diesel system. Conversely, ozone layer depletion potential, human toxicity, acidification, and eutrophication potential are greater in the diesel system. Overall, the grid electricity system exhibits elevated values across a greater number of impact indicators, suggesting a comparatively higher environmental impact.

Fig 10 presents a comparative analysis of impact category trends across the two systems. The figure reveals that both systems follow a similar trend, with marine aquatic ecotoxicity, abiotic depletion (fossil fuels), human toxicity, global warming potential, and acidification potential emerging as the most substantial contributors to environmental impact in both cases. This similarity suggests that, despite differences in energy sources, certain impact categories consistently exert a dominant influence on the overall environmental footprint of water extraction systems.

A quantitative comparison of the two systems was subsequently performed, utilizing the environmental composite index as the metric for evaluation. These results are also presented in Fig 11. The diesel system exhibited a mean ECI value of $3.68 \times 10^{-4}$ nPt, whereas the grid electricity system yielded a mean ECI of $5.77 \times 10^{-4}$ nPt. This analysis indicates that the environmental impact associated with well water extraction using grid electricity is 56.79% greater than that observed for the diesel-powered system.

Pradeleix et al. [29] employed LCA to evaluate the environmental impacts associated with diverse groundwater pumping systems in semi-arid Tunisia. Their findings indicated that diesel-powered pumping systems exhibit greater environmental burdens than electrically powered systems when the electricity source is natural gas and diesel pump efficiency is suboptimal. Conversely, the study demonstrated that under conditions of coal-derived electricity generation and diesel pump efficiencies exceeding 12%, diesel-powered systems become the environmentally preferable option.

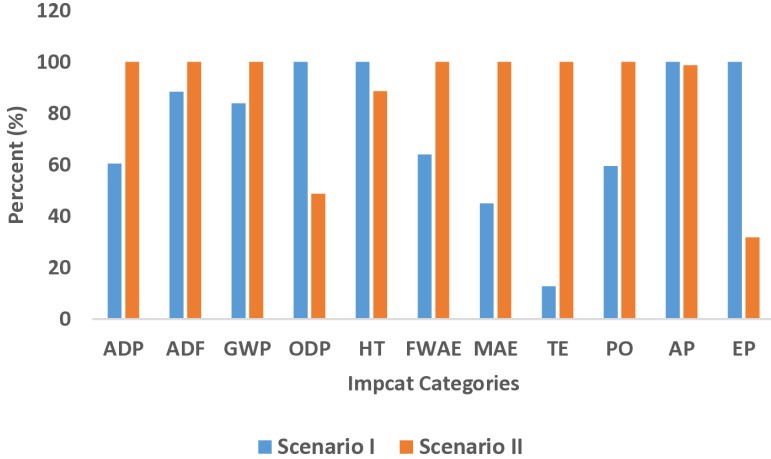

**Fig 9. Comparative diagram of impact categories in diesel-powered and grid-powered systems.**

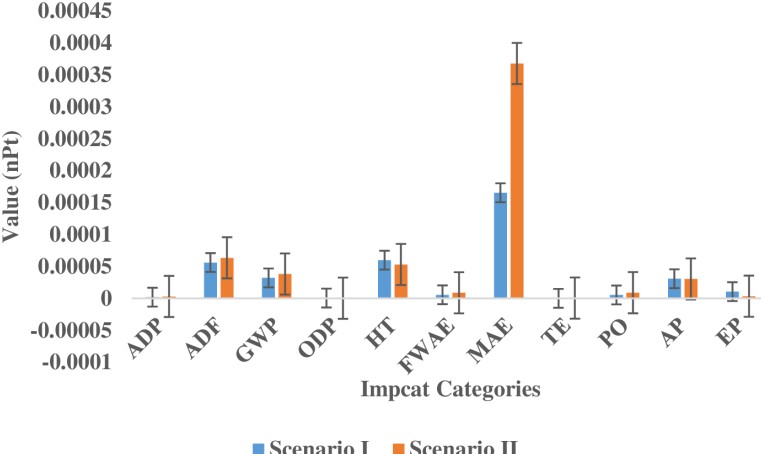

**Fig 10. Comparative analysis of impact category trends in diesel-powered and grid-powered systems.** The figure illustrates the relative contributions of different impact categories (in nPt) to the overall environmental burden for both diesel-powered and grid-powered systems. Key trends include the dominance of marine aquatic ecotoxicity in both systems, with higher impacts in grid-powered systems due to fossil fuel-based electricity generation.

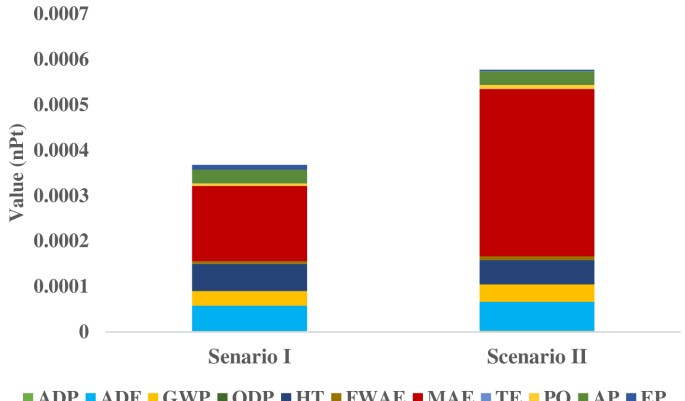

**Fig 11. Comparison of environmental composite index values for diesel-powered and grid-powered systems.** The figure shows the aggregated environmental burden (in nPt) per cubic meter of water extracted for both systems. The grid-powered system exhibits a 56.79% higher ECI compared to the diesel-powered system, primarily due to fossil fuel-based electricity generation.

It is important to acknowledge the prevalence of suboptimal operating efficiencies in diesel-powered pumping systems within Iranian wells, largely attributable to engine degradation. While the diesel system exhibited a lower ECI than the grid electricity system, this value remains suboptimal and amenable to improvement through enhanced efficiency measures. Modernization of wellhead systems and the implementation of advanced equipment are anticipated to positively impact environmental performance indicators, resulting in a more favorable overall ECI. However, dynamic and kinematic analyses of diesel engine-related systems indicate that improving design and optimizing operational conditions can lead to reduced energy consumption and lower greenhouse gas emissions [51,52]. This insight underscores the importance of evaluating both energy efficiency and environmental impacts when comparing diesel and electric water pumping systems for agricultural irrigation. From a sustainability standpoint, further investigation of the economic and social implications of diesel-powered water extraction, relative to grid electricity-powered extraction, is recommended. Furthermore, a comprehensive assessment of water extraction systems driven by renewable energy sources, including solar, wind, and biogas

technologies, is warranted to evaluate their viability and environmental advantages. Recent studies on hybrid renewable energy systems have demonstrated that optimizing system design and parameters can significantly improve stability and energy capture efficiency [53]. These findings underscore the importance of system optimization in renewable energy applications, which is equally relevant when comparing diesel and electric water pumping systems for agricultural irrigation. By applying similar optimization principles, future research could enhance the performance and sustainability of water extraction systems, particularly those powered by renewable energy sources.

### 3.4. Effects of local and technical parameters on environmental impacts of irrigation pumping systems

The findings of this study suggest that, for well water extraction in Iran, diesel-fueled pumping systems represent a more environmentally sustainable option compared to grid-connected electric systems. This can be attributed to several factors, including the direct energy conversion in diesel systems, which eliminates transmission and distribution losses associated with grid electricity. In contrast, grid-connected systems in Iran suffer from inefficiencies in power generation, transmission, and distribution, exacerbated by the country's heavy reliance on fossil fuels for electricity generation (approximately 94% of Iran's electricity is generated from fossil fuels). These factors collectively contribute to the higher environmental burden of grid-connected systems. The existing literature presents divergent perspectives on the most environmentally favorable energy source for agricultural irrigation water pumping. While certain studies (e.g., [17] and [27]) highlight the advantages of diesel-powered systems over grid-connected alternatives, others (e.g., [14] and [26]) report opposing conclusions. These discrepancies can be partly attributed to variations in electricity generation sources. For instance, the findings of this study align with those of Pradeleix et al. [29] for Tunisia, where diesel systems were also found to have a lower environmental impact compared to grid electricity. This similarity can be explained by the shared characteristic of fossil fuel-dominated electricity grids in both countries, which result in high carbon intensity and significant transmission losses. However, the findings contrast with those of El-Gafy & El-Bably [14] for Egypt, where grid electricity was reported to be more environmentally favorable. This discrepancy can be attributed to differences in the energy mix and grid efficiency between the two countries. Furthermore, the substantial impact of primary energy sources on the environmental performance of electrically powered water pumping systems has been emphasized in prior research [28,29].

$CO_2$ emissions from electricity generation are influenced by several key factors, including economic growth, population size, electricity and energy intensity, the generation fuel mix, power plant efficiency, electricity trade, and environmental policies [54]. Table 8 provides a comparative analysis of the energy source compositions utilized by electricity-generating power plants in Iran during the year 2022. This analysis contrasts Iran's energy sources with those employed by several regions.

Table 8. Comparative analysis of electricity generation sources in Iran and selected regions [18].

| Region/Country | Source of electricity generation | | | | | | | | |
| --- | --- | --- | --- | --- | --- | --- | --- | --- | --- |
| | Fossil Sources (%) | | | Clean Sources (%) | | | | | |
| | Coal | Oil | Natural gas | Biofuels & Waste | Nuclear | Hydro | Geothermal | Solar | Wind |
| Iran | 0.19 | 13.54 | 80.21 | 0.01 | 2.20 | 3.43 | 0 | 0.20 | 0.22 |
| | 93.94 | | | 6.06 | | | | | |
| Middle East | 0.07 | 25.67 | 69.32 | 0 | 2.14 | 1.25 | 0 | 1.24 | 0.31 |
| | 95.06 | | | 4.94 | | | | | |
| Eurasia | 18.28 | 0.77 | 46.32 | 0.28 | 15.5 | 17.90 | 0.03 | 0.38 | 0.55 |
| | 65.37 | | | 34.64 | | | | | |
| Europe | 17.10 | 1.55 | 20.95 | 5.98 | 18.66 | 14.8 | 0.58 | 6.36 | 13.84 |
| | 39.60 | | | 60.22 | | | | | |
| North America | 17.55 | 1.32 | 36.59 | 1.41 | 16.42 | 13 | 0.43 | 3.89 | 9.05 |
| | 55.46 | | | 44.20 | | | | | |

Table 8 illustrates the dominance of fossil fuels in Iran's power generation, accounting for approximately 94% of the total energy supply. Conversely, renewable energy sources and nuclear power contribute minimally, comprising only about 6%. This heavy reliance on fossil fuels results in a comparatively higher environmental impact associated with electricity consumption in Iran than in some other nations. As detailed in Table 8, European countries exhibit a significantly different energy profile, with over 60% of electricity generation derived from clean energy sources and only approximately 40% from fossil fuels. The European Union's greater success in implementing and utilizing nuclear energy and renewable energy sources has been previously documented [55].

To facilitate a comprehensive analysis and comparative assessment of carbon dioxide emissions across nations, the aggregate carbon intensity (ACI) index can be employed [56]. This metric quantifies the amount of carbon dioxide released per unit of electricity generated (kilowatt-hour). A lower ACI value signifies a more favorable environmental performance. In 2013, Iran's ACI was measured at 0.5444 $kgCO_2$/kWh. However, this value remains comparatively elevated within the global context, positioning Iran 15th in terms of ACI magnitude [56]. While developed economies have demonstrated a downward trend in carbon dioxide emissions from electricity generation in recent years, developing and emerging economies continue to exhibit an upward trajectory [57]. In Iran, renewable energy sources, such as wind and solar photovoltaic technologies, have gradually developed, albeit to a limited extent [18]. However, in numerous instances, these renewable sources have not displaced fossil fuels; rather, they have primarily been employed to meet the growing electricity demand. It is essential to note that achieving a reduction in ACI through renewable energy development requires the substitution of fossil fuels with renewable sources [55].

A sensitivity analysis was conducted to evaluate the impact of varying energy mixes on environmental outcomes. Scenarios with 20%, 40%, and 60% renewable energy were simulated, resulting in GWP reductions of 15%, 35%, and 55%, respectively, compared to the current grid mix. These results highlight the environmental benefits of transitioning to a cleaner energy mix.

A further consideration pertains to energy conversion efficiency. In diesel-powered systems, fuel is directly converted to mechanical work, eliminating transmission and distribution losses. Conversely, grid-connected electrical systems experience losses due to resistance in transmission lines and distribution infrastructure [24,58].

Furthermore, the efficiency of electric pumps warrants consideration. Pump efficiency significantly influences system performance, and the prevalence of non-standard, low-efficiency pumps in Iranian well water extraction is a notable concern. While a comprehensive assessment of irrigation pump status in Iran is lacking, research conducted in comparable contexts, such as the study by Shelar et al. [59] in Maharashtra, India, suggests analogous conditions. Their investigation into end-use efficiency of irrigation pumps, focusing on demand-side electricity management in rural areas, revealed that 75% of agricultural pumps were unbranded, resulting in diminished efficiency and financial detriments for both farmers and power providers. Similarly, the performance and efficiency of diesel engines play a crucial role in determining their environmental impact, particularly in applications such as agricultural water pumping. Studies have shown that optimizing diesel engine components can enhance engine stability and reduce mechanical losses, ultimately contributing to improved fuel efficiency and lower emissions [60].

The primary drivers of environmental impacts in grid-connected electricity systems are closely tied to the electricity generation phase, which in Iran is heavily reliant on fossil fuels. This reliance results in significant emissions of greenhouse gases and other pollutants, particularly in impact categories such as GWP, ADF, and MAE. Additionally, transmission and distribution losses further exacerbate the environmental burden of grid-connected systems. The composition of the electricity grid, dominated by fossil fuels, plays a critical role in determining the overall environmental impact. If the grid were to incorporate a higher proportion of renewable energy sources, these impacts could be significantly reduced. Furthermore, the inefficiencies of Iran's fossil fuel-based power plants contribute to the environmental burden, highlighting the need for improvements in power generation efficiency and the adoption of cleaner technologies.

The study proposes several strategies for environmental mitigation to reduce the impacts of well water extraction systems. These include the integration of renewable energy sources (e.g., solar and wind) to power water extraction, modernization of power generation infrastructure to improve efficiency, and enhancement of pump efficiency through technological upgrades. Policy interventions, such as subsidies for renewable energy systems and penalties for inefficient pumps, can further drive positive change. Additionally, improved water management practices, such as drip irrigation and precision agriculture, can reduce water demand and energy consumption. Finally, investing in research and development to explore innovative solutions for sustainable water extraction and energy use is essential for long-term environmental benefits. These strategies, if implemented effectively, can significantly reduce the environmental impacts of well water extraction systems while maintaining agricultural productivity.

Transitioning to renewable energy systems, such as solar-powered water pumps, presents a viable solution for reducing the environmental impacts of agricultural water extraction in Iran. Studies like Guno [32] demonstrate the economic and environmental benefits of solar irrigation systems, including significant fuel cost savings and reductions in greenhouse gas emissions. Similarly, Terang & Baruah [26] highlight the role of targeted subsidies and supportive policies in promoting the adoption of solar photovoltaic water pumps (SPVWP), particularly in regions with high solar potential. Despite challenges such as high initial investment costs and the need for technical training, the social acceptability of solar pumps among farmers suggests a positive outlook for renewable energy adoption. To facilitate this transition, policies that prioritize subsidies, public-private partnerships, and capacity-building initiatives are recommended.

## 4. Conclusion

This study provided a comprehensive environmental impact assessment of water extraction systems powered by diesel and grid electricity within the specific context of Iran's agricultural sector. The life cycle analysis (LCA) approach was employed to evaluate various environmental impact categories, including global warming potential (GWP), abiotic depletion, human toxicity, ecotoxicity, acidification, and eutrophication.

The findings indicate that diesel-powered water extraction systems generally exhibit lower environmental impacts compared to grid-connected electric systems, particularly in terms of global warming potential and overall environmental composite index (ECI). This advantage is largely attributable to Iran's heavy reliance on fossil fuel-based electricity generation, which significantly increases the carbon intensity of grid-supplied electricity. Specifically, the diesel system demonstrated a 56.79% lower ECI than the grid electricity system, underscoring its relative environmental advantage under current conditions.

However, it is crucial to emphasize that this advantage is context-specific and applies primarily to regions with energy grids heavily reliant on fossil fuels. In regions where grid electricity is sourced predominantly from renewable energy, electric-powered systems may offer superior environmental performance. For regions with energy profiles similar to Iran's-characterized by a high dependence on fossil fuels for electricity generation-the findings suggest that diesel-powered systems may present a more environmentally favorable option under current technical and operational conditions. However, this does not imply universal applicability, as the environmental performance of these systems is highly sensitive to the energy mix of the grid and the efficiency of the technologies employed.

Furthermore, the study highlighted the critical influence of technical parameters, such as the efficiency of pumping equipment and transmission losses in electric systems, on the overall environmental impact. While diesel systems showed lower emissions in this context, they are not without environmental drawbacks, particularly concerning human toxicity, ozone layer depletion, and acidification potential. Moreover, the suboptimal operational efficiency of diesel pumps in Iran presents opportunities for environmental improvement through modernization and maintenance.

The present study is subject to certain limitations that warrant acknowledgement. Firstly, the scope of this investigation did not encompass renewable energy systems, such as solar photovoltaic or wind power, which possess the potential to

yield substantial environmental advantages. Subsequent research endeavors should prioritize the exploration of these alternative energy sources, with particular emphasis on hybrid configurations (e.g., solar-diesel), to ascertain more sustainable methodologies for agricultural irrigation practices.

Secondly, the analysis incorporated assumptions regarding pump efficiency, particularly concerning prevalent, older-generation diesel engines utilized within Iran. Future investigations should incorporate empirical field measurements of pump performance and explore strategies for efficiency enhancement to refine the accuracy of assessments.

Thirdly, the findings of this study are context-dependent, being specific to the prevailing fossil fuel-centric electricity grid of Iran. Consequently, the generalizability of these results to regions characterized by divergent energy portfolios may be constrained. Future research should examine water extraction systems across a spectrum of diverse energy contexts to facilitate the development of broader and more universally applicable insights.

Finally, the current study maintained a singular focus on environmental impacts. Future research initiatives should adopt a more holistic perspective by integrating economic and social dimensions to cultivate a more comprehensive understanding of sustainability within the context of agricultural water extraction.

## Supporting information

**S1 File. Dataset used in the LCA analysis, including impact categories, weighted results, and composite scores for diesel and grid-powered irrigation systems.**
(ZIP)

## Author contributions

**Conceptualization:** Majid Namdari.

**Data curation:** Majid Namdari.

**Formal analysis:** Majid Namdari.

**Funding acquisition:** Majid Namdari.

**Investigation:** Majid Namdari.

**Methodology:** Majid Namdari.

**Project administration:** Majid Namdari.

**Resources:** Majid Namdari.

**Software:** Majid Namdari.

**Supervision:** Majid Namdari.

**Validation:** Majid Namdari.

**Visualization:** Majid Namdari.

**Writing – original draft:** Majid Namdari.

**Writing – review & editing:** Majid Namdari.

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
