## [Decision Letter · Decision Letter 0]

12 Mar 2025

Dear Dr. Namdari,

We look forward to receiving your revised manuscript.

Kind regards,

Morteza Taki, Ph.D

Academic Editor

PLOS ONE

Reviewers' comments:

Reviewer's Responses to Questions

**Comments to the Author**

1. Is the manuscript technically sound, and do the data support the conclusions?

Reviewer #1: Partly

Reviewer #2: Partly

2. Has the statistical analysis been performed appropriately and rigorously?

Reviewer #1: No

Reviewer #2: I Don't Know

3. Have the authors made all data underlying the findings in their manuscript fully available?

Reviewer #1: No

Reviewer #2: Yes

4. Is the manuscript presented in an intelligible fashion and written in standard English?

Reviewer #1: No

Reviewer #2: No

Reviewer #1: What are the key environmental indicators evaluated in the study using Life Cycle Assessment (LCA) methodology?

How does the Environmental Composite Index (ECI) of diesel-powered well water extraction systems compare to grid-connected electricity systems?

What percentage higher environmental burden is associated with grid electricity systems compared to diesel-powered systems?

What are the primary drivers of environmental impacts in grid-connected electricity systems according to the study?

Which environmental impact categories do grid electricity systems exhibit higher impacts in compared to diesel systems?

In which impact categories do diesel systems demonstrate greater impacts compared to grid electricity systems?

How do local and technical factors, such as energy source composition and power plant efficiency, influence environmental outcomes in well water extraction systems?

Revise the text by native English editor

What strategies for environmental mitigation are suggested by the study to reduce the impacts of well water extraction systems?

The below references can be used in the introduction section to improve it:

Guo, H., Li, Y., Zhao, B., Guo, Y., Xie, Z., Morina, A.,... Lu, X. (2024). Transient lubrication of floating bush coupled with dynamics and kinematics of cam-roller in fuel supply mechanism of diesel engine. Physics of Fluids, 36(12), 123103. doi: https://doi.org/10.1063/5.0232226

Chen, L., Cui, B., Zhang, C., Hu, X., Wang, Y., Li, G.,... Liu, L. (2024). Impacts of Fuel Stage Ratio on the Morphological and Nanostructural Characteristics of Soot Emissions from a Twin Annular Premixing Swirler Combustor. Environmental Science & Technology, 58(24), 10558-10566. doi: https://doi.org/10.1021/acs.est.4c03478

Chen, L., Cao, Y., Hu, X., Zhang, B., Chen, X., Cui, B.,... Xu, Z. (2025). Raman spectral optimization for soot particles: A comparative analysis of fitting models and machine learning enhanced characterization in combustion systems. Building and Environment, 271, 112600. doi: https://doi.org/10.1016/j.buildenv.2025.112600

Qiu, Z., Chen, R., Gan, X., & Wu, C. (2024). Torsional damper design for diesel engine: theory and application. Physica Scripta, 99(12), 125214. doi: 10.1088/1402-4896/ad8af8

Zhang, Z., Bu, Y., Wu, H., Wu, L., & Cui, L. (2023). Parametric study of the effects of clump weights on the performance of a novel wind-wave hybrid system. Renewable Energy, 219(Part 1), 119464. doi: https://doi.org/10.1016/j.renene.2023.119464

Reviewer #2: 1. Introduction & Literature Review

The introduction should better contextualize the study within recent global trends in renewable energy adoption for irrigation (e.g., solar/wind). Expand the literature review to include studies from regions with similar energy grids (e.g., Middle East) to strengthen the rationale for comparing diesel and grid electricity in Iran. I recommend to apply the below reference in this section:

Yao, Y., Sheng, H., Luo, Y., He, M., Li, X., Zhang, H., ... & An, L. (2014). Optimization of anaerobic co-digestion of Solidago canadensis L. biomass and cattle slurry. Energy, 78, 122-127.

Yao, Y., Bergeron, A. D., & Davaritouchaee, M. (2018). Methane recovery from anaerobic digestion of urea-pretreated wheat straw. Renewable energy, 115, 139-148.

Deng, Y., Qiu, L., Shao, Y., & Yao, Y. (2019). Process modeling and optimization of anaerobic Co-digestion of peanut hulls and swine manure using response surface methodology. Energy & Fuels, 33(11), 11021-11033.

Qiu, L., Deng, Y. F., Wang, F., Davaritouchaee, M., & Yao, Y. Q. (2019). A review on biochar-mediated anaerobic digestion with enhanced methane recovery. Renewable and Sustainable Energy Reviews, 115, 109373.

Wu, H., Li, A., Zhang, H., Li, S., Yang, C., Lv, H., & Yao, Y. (2024). Microbial mechanisms for higher hydrogen production in anaerobic digestion at constant temperature versus gradient heating. Microbiome, 12(1), 170.

Zhang, Y., Xu, L., Wang, J., Pan, H., Dou, M., Teng, Y., ... & Wang, M. (2024). Bagasse-based porous flower-like MoS2/carbon composites for efficient microwave absorption. Carbon Letters, 1-16.

Yang, Z. Q., & Zhu, H. L. (2023). Study on the effect of carbon nanotubes on the microstructure and anti-carbonation properties of cement-based materials. Journal of Functional Materials, 54(08), 8217-8227.

2. Methodology: LCA Approach & Data Representativeness

Clarify why the CML methodology and SimaPro software were chosen over other LCA frameworks (e.g., ReCiPe). Justify the sample size (100 wells) and geographic distribution (Tehran, Hamadan, etc.) to ensure representativeness. Address potential biases in data collection (e.g., self-reported diesel use vs. metered electricity).

3. System Boundaries & Energy Mix Assumptions

Explicitly define system boundaries for both scenarios (e.g., inclusion/exclusion of transmission losses, pump maintenance). Discuss how Iran’s electricity grid composition (94% fossil fuels, per Table 8) was modeled, and sensitivity analysis for varying energy mixes (e.g., renewables) should be included.

4. Results Presentation & Figure Clarity

Revise Figures 5–11 for clarity (e.g., label axes, simplify color schemes). Ensure consistency in units (e.g., nPt vs. kg CO₂-eq) and provide error bars/uncertainty ranges for impact categories. Highlight key drivers (e.g., marine ecotoxicity in diesel systems) in figure captions.

5. Interpretation of Key Findings

Strengthen the discussion of why diesel outperforms grid electricity in Iran (e.g., direct energy conversion vs. grid inefficiencies). Compare results with studies in similar contexts (e.g., Pradeleix et al. 2015 for Tunisia) and explain discrepancies (e.g., El-Gafy & El-Bably 2016 for Egypt).

6. Limitations & Future Research

Expand the limitations section to address:

• Lack of renewable energy systems in the analysis.

• Assumptions about pump efficiency (e.g., outdated diesel engines).

• Geographic specificity of Iran’s grid. Suggest future studies on hybrid systems (e.g., solar-diesel).

7. Conclusion Overgeneralization

Revise the conclusion to emphasize that the diesel advantage is context-specific to Iran’s fossil-reliant grid. Avoid implying universal applicability; instead, frame recommendations for regions with similar energy profiles.

8. Renewable Energy Integration

Discuss policy implications for transitioning to renewables (e.g., solar pumps) given Iran’s high solar potential. Reference studies like Guno (2024) and Terang & Baruah (2023) to support this.

9. Language & Technical Precision

Proofread for grammatical errors (e.g., "electricity system’s" vs. "electricity systems"). Use consistent terminology (e.g., "abiotic depletion" vs. "resource depletion") and define acronyms (e.g., ECI, FU) at first use.

10. Title & Keyword Specificity

Revise the title to reflect Iran’s context and key findings: “Comparative LCA of Diesel and Grid Electricity for Irrigation in Iran: Why Diesel Outperforms in Fossil-Reliant Grids.”

Add keywords like “Iran,” “groundwater pumping,” and “renewable energy transition.”

**Do you want your identity to be public for this peer review?** For information about this choice, including consent withdrawal, please see our Privacy Policy

Reviewer #1: No

Reviewer #2: No

---

## [Author Response · Author response to Decision Letter 1]

8 Apr 2025

Dear Editors and Reviewers,

Thank you very much for your valuable feedback and constructive comments on our manuscript titled "Comparative LCA of diesel and grid electricity for agricultural irrigation in Iran: why diesel outperforms in fossil-reliant grids" We viewers’ comments and outline the changes made to the manuscript.

Reviewer #1:

# Comment 1: What are the key environmental indicators evaluated in the study using Life Cycle Assessment (LCA) methodology?

Response:

Thank you for your valuable comment. In response to your question regarding the key environmental indicators evaluated in the study, we have added a detailed explanation in the Materials and Methods section of the manuscript. Specifically, we have included a new paragraph (highlighted in yellow) that lists and describes the environmental impact categories assessed using the CML methodology. These categories include:

• Abiotic Depletion (ADP)

• Abiotic Depletion (Fossil Fuels) (ADF)

• Global Warming Potential (GWP)

• Ozone Layer Depletion Potential (ODP)

• Human Toxicity (HT)

• Freshwater Aquatic Ecotoxicity (FWAE)

• Marine Aquatic Ecotoxicity (MAE)

• Terrestrial Ecotoxicity (TE)

• Photochemical Oxidation (PO)

• Acidification Potential (AP)

• Eutrophication Potential (EP)

We believe this addition provides a clearer and more comprehensive explanation of the environmental indicators evaluated in our study, addressing your concern directly. Please find the revised text in the updated manuscript.

# Comment 2: How does the Environmental Composite Index (ECI) of diesel-powered well water extraction systems compare to grid-connected electricity systems?

Response:greatly appreciate the time and effort you have dedicated to reviewing our work. We have carefully considered all the comments and suggestions, and have made the necessary revisions to improve the quality of the manuscript. Below, we provide a point-by-point response to the re

Dear Reviewer, Thank you for your valuable comment regarding the comparison of the Environmental Composite Index (ECI) between diesel-powered and grid-connected electricity systems. We would like to highlight that this comparison and its underlying reasons are already discussed in detail in the Results and Discussion section of the manuscript. Specifically, please refer to the following parts:

Section 3.3 (Comparative Analysis of Environmental Impacts: Diesel vs. Electricity-Powered Systems): This section provides a comprehensive comparison of the ECI values for both systems, highlighting that the diesel-powered system has a lower ECI (3.68 × 10-4 nPt) compared to the grid-connected system (5.77 × 10-4 nPt), representing a 56.79% lower environmental burden for diesel systems.

Section 3.4 (Effects of Local and Technical Parameters on Environmental Impacts of Irrigation Pumping Systems): This section explains the reasons for this difference, including the heavy reliance on fossil fuels in Iran's electricity grid, the absence of transmission losses in diesel systems, and the inefficiencies of Iran's power generation infrastructure.

We believe that these sections address your question thoroughly. However, to make this information more accessible, we have highlighted the relevant paragraphs in yellow in the revised manuscript.

Thank you again for your insightful comment, which has helped us ensure that the key findings of our study are clearly communicated.

# Comment 3: What percentage higher environmental burden is associated with grid electricity systems compared to diesel-powered systems?

Response:

The grid-connected electricity system exhibits a 56.79% higher environmental burden compared to the diesel-powered system, as measured by the Environmental Composite Index (ECI). Specifically, the ECI for the grid-connected system is 5.77 × 10-4 nPt, while the ECI for the diesel-powered system is 3.68 × 10-4 nPt. This difference highlights the significant environmental impact associated with grid electricity, primarily due to Iran's heavy reliance on fossil fuels for electricity generation and the inefficiencies in power transmission and distribution.

# Comment 4: What are the primary drivers of environmental impacts in grid-connected electricity systems according to the study?

Response:

Thank you for your valuable comment regarding the primary drivers of environmental impacts in grid-connected electricity systems. In response to your question, we have added a detailed explanation in the Results and Discussion section of the manuscript. Specifically, we have included a new paragraph (highlighted in yellow) that identifies and discusses the key factors contributing to the environmental impacts of grid-connected systems. These factors include:

• The heavy reliance on fossil fuels for electricity generation in Iran, which results in significant emissions of greenhouse gases (GHGs) and other pollutants.

• Transmission and distribution losses, which exacerbate the environmental burden.

• The composition of the electricity grid, dominated by fossil fuels, and its influence on impact categories such as Global Warming Potential (GWP), Abiotic Depletion (Fossil Fuels) (ADF), and Marine Aquatic Ecotoxicity (MAE).

• The inefficiencies of Iran's fossil fuel-based power plants, which further contribute to the environmental burden.

We believe this addition provides a clearer and more comprehensive explanation of the primary drivers of environmental impacts in grid-connected systems, addressing your concern directly. Please find the revised text in the updated manuscript.

Thank you again for your insightful comment, which has helped improve the clarity of our manuscript.

# Comment 5: Which environmental impact categories do grid electricity systems exhibit higher impacts in compared to diesel systems?

In which impact categories do diesel systems demonstrate greater impacts compared to grid electricity systems?

Response:

Thank you for your valuable comment regarding the comparison of environmental impact categories between grid-connected electricity systems and diesel-powered systems. We would like to highlight that this comparison is already discussed in detail in the Results and Discussion section of the manuscript. Specifically, please refer to the following parts:

Section 3.3 (Comparative Analysis of Environmental Impacts: Diesel vs. Electricity-Powered Systems): This section provides a comprehensive comparison of the environmental impact categories for both systems, highlighting the categories where grid-connected systems exhibit higher impacts (e.g., Abiotic Depletion, Global Warming Potential, and Marine Aquatic Ecotoxicity) and those where diesel systems demonstrate greater impacts (e.g., Ozone Layer Depletion, Human Toxicity, and Acidification Potential).

We believe that these sections address your question thoroughly. However, to make this information more accessible, we have highlighted the relevant paragraphs in yellow in the revised manuscript.

Thank you again for your insightful comment, which has helped us ensure that the key findings of our study are clearly communicated.

# Comment 6: How do local and technical factors, such as energy source composition and power plant efficiency, influence environmental outcomes in well water extraction systems?

Response:

Thank you for your valuable comment regarding the influence of local and technical factors on the environmental outcomes of well water extraction systems.

Section 3.4 (Effects of Local and Technical Parameters on Environmental Impacts of Irrigation Pumping Systems): This section provides a comprehensive analysis of how factors such as energy source composition, power plant efficiency, transmission and distribution losses, and pump efficiency influence the environmental performance of both diesel-powered and grid-connected systems.

we have highlighted the relevant paragraphs in yellow in the revised manuscript.

Thank you again for your insightful comment

# Comment 7: Revise the text by native English editor

Response:

Thank you for your valuable comment regarding the need for language editing. We acknowledge the importance of ensuring that the manuscript is written in clear and fluent English. In response to your suggestion, we have engaged a professional native English editor to revise the entire manuscript. The editor has carefully reviewed the text to improve grammar, sentence structure, and overall readability while ensuring that the scientific content remains accurate and unchanged.

We believe that these revisions have significantly enhanced the quality of the manuscript. The edited version is now ready for your review, and we hope that it meets the journal's standards for language and clarity.

# Comment 8: What strategies for environmental mitigation are suggested by the study to reduce the impacts of well water extraction systems?

Response:

Thank you for your valuable comment regarding the strategies for environmental mitigation to reduce the impacts of well water extraction systems. In response to your question, we have added a detailed explanation in the Results and Discussion section of the manuscript. Specifically, we have included a new paragraph (highlighted in yellow) that outlines the key mitigation strategies proposed by the study. These strategies include:

Integration of renewable energy sources (e.g., solar and wind) to power water extraction systems.

• Modernization of power generation infrastructure to improve efficiency.

• Enhancement of pump efficiency through technological upgrades.

• Policy interventions, such as subsidies for renewable energy systems and penalties for inefficient pumps.

• Improved water management practices, such as drip irrigation and precision agriculture.

• Investment in research and development to explore innovative solutions for sustainable water extraction and energy use.

We believe this addition provides a clearer and more comprehensive explanation of the proposed mitigation strategies, addressing your concern directly. Please find the revised text highlighted in yellow in the updated manuscript.

# Comment 9: The below references can be used in the introduction section to improve it:

• Guo, H., Li, Y., Zhao, B., Guo, Y., Xie, Z., Morina, A.,... Lu, X. (2024). Transient lubrication of floating bush coupled with dynamics and kinematics of cam-roller in fuel supply mechanism of diesel engine. Physics of Fluids, 36(12), 123103. doi: https://doi.org/10.1063/5.0232226

• Chen, L., Cui, B., Zhang, C., Hu, X., Wang, Y., Li, G.,... Liu, L. (2024). Impacts of Fuel Stage Ratio on the Morphological and Nanostructural Characteristics of Soot Emissions from a Twin Annular Premixing Swirler Combustor. Environmental Science & Technology, 58(24), 10558-10566. doi: https://doi.org/10.1021/acs.est.4c03478

• Chen, L., Cao, Y., Hu, X., Zhang, B., Chen, X., Cui, B.,... Xu, Z. (2025). Raman spectral optimization for soot particles: A comparative analysis of fitting models and machine learning enhanced characterization in combustion systems. Building and Environment, 271, 112600. doi: https://doi.org/10.1016/j.buildenv.2025.112600

• Qiu, Z., Chen, R., Gan, X., & Wu, C. (2024). Torsional damper design for diesel engine: theory and application. Physica Scripta, 99(12), 125214. doi: 10.1088/1402-4896/ad8af8

• Zhang, Z., Bu, Y., Wu, H., Wu, L., & Cui, L. (2023). Parametric study of the effects of clump weights on the performance of a novel wind-wave hybrid system. Renewable Energy, 219(Part 1), 119464. doi: https://doi.org/10.1016/j.renene.2023.119464

Response:

Thank you for your valuable suggestion regarding the inclusion of additional references to improve the introduction section. We appreciate your recommendations and have carefully reviewed the suggested references. In response to your comment, we have incorporated these references in the manuscript. We believe that these additions have enhanced the quality and depth of the manuscript.

Reviewer #2:

# Comment 1: Introduction & Literature Review

The introduction should better contextualize the study within recent global trends in renewable energy adoption for irrigation (e.g., solar/wind). Expand the literature review to include studies from regions with similar energy grids (e.g., Middle East) to strengthen the rationale for comparing diesel and grid electricity in Iran. I recommend to apply the below reference in this section:

1. Yao, Y., Sheng, H., Luo, Y., He, M., Li, X., Zhang, H., ... & An, L. (2014). Optimization of anaerobic co-digestion of Solidago canadensis L. biomass and cattle slurry. Energy, 78, 122-127.

2. Yao, Y., Bergeron, A. D., & Davaritouchaee, M. (2018). Methane recovery from anaerobic digestion of urea-pretreated wheat straw. Renewable energy, 115, 139-148.

3. Deng, Y., Qiu, L., Shao, Y., & Yao, Y. (2019). Process modeling and optimization of anaerobic Co-digestion of peanut hulls and swine manure using response surface methodology. Energy & Fuels, 33(11), 11021-11033.

4. Qiu, L., Deng, Y. F., Wang, F., Davaritouchaee, M., & Yao, Y. Q. (2019). A review on biochar-mediated anaerobic digestion with enhanced methane recovery. Renewable and Sustainable Energy Reviews, 115, 109373.

5. Wu, H., Li, A., Zhang, H., Li, S., Yang, C., Lv, H., & Yao, Y. (2024). Microbial mechanisms for higher hydrogen production in anaerobic digestion at constant temperature versus gradient heating. Microbiome, 12(1), 170.

6. Zhang, Y., Xu, L., Wang, J., Pan, H., Dou, M., Teng, Y., ... & Wang, M. (2024). Bagasse-based porous flower-like MoS2/carbon composites for efficient microwave absorption. Carbon Letters, 1-16.

7. Yang, Z. Q., & Zhu, H. L. (2023). Study on the effect of carbon nanotubes on the microstructure and anti-carbonation properties of cement-based materials. Journal of Functional Materials, 54(08), 8217-8227.

Response:

Thank you for your valuable comment regarding the need to better contextualize the study within recent global trends in renewable energy adoption for irrigation and to expand the literature review to include studies from regions with similar energy grids. In response to your suggestion, we have revised the Introduction section. We appreciate your recommendations and have carefully reviewed the suggested references. In response to your comment, we have incorporated these references in the manuscript.

# Comment 2: Methodology: LCA Approach & Data Representativeness

Clarify why the CML methodology and SimaPro software were chosen over other LCA frameworks (e.g., ReCiPe). Justify the sample size (100 wells) and geographic distribution (Tehran, Hamadan, etc.) to ensure representativeness. Address potential biases in data collection (e.g., self-reported diesel use vs. metered electricity).

Response:

Thank you for your valuable comments regarding the methodology of our study. Below, we address each of your concerns in detail:

1- Justification for Choosing the CML Methodology and SimaPro Software:

The CML methodology was selected for this study due to its objectivity and international recognition in environmental impact assessment. Unlike other frameworks such as ReCiPe, which incorporate subjective weighting factors based on societal preferences, the CML methodology relies on scientific and objective criteria for impact assessment. This makes it particularly suitable for comparative studies like ours, where the goal is to provide an unbiased evaluation of environmental impacts (Dekamin et al., 2018; Rasooli et al., 2023). Additionally, the CML methodology is well-documented and widely used in LCA studies, ensuring consistency and comparability with other research.

The SimaPro software was chosen because it is one of the most comprehensive and widely used tools for LCA studies. It supports the CML methodology and provides a robust platform for modeling complex systems, such as diesel-powered and grid-connected irrigation systems. SimaPro's extensive database and flexibility in handling large datasets make it ideal for this study.

2- Justification for Sample Size and Geographic Distribution:

The sample size of 100 wells was determined based on the need to achieve a representative and statistically robust dataset while considering practical constraints such as data availability and accessibility. This sample size is sufficient to capture the variability i

---

## [Decision Letter · Decision Letter 1]

14 Apr 2025

Dear Dr. Namdari,

Thank you for submitting your manuscript to PLOS ONE. After careful consideration, we feel that it has merit but does not fully meet PLOS ONE’s publication criteria as it currently stands. Therefore, we invite you to submit a revised version of the manuscript that addresses the points raised during the review process.

We look forward to receiving your revised manuscript.

Kind regards,

Morteza Taki, Ph.D

Academic Editor

PLOS ONE

Journal Requirements:

Additional Editor Comments:

Dear Dr Namdari

The reviewers have generally accepted your paper, but it requires some minor revisions. Please review the comments and make the necessary edits to the paper.

Reviewers' comments:

Reviewer's Responses to Questions

**Comments to the Author**

Reviewer #1: All comments have been addressed

Reviewer #2: All comments have been addressed

2. Is the manuscript technically sound, and do the data support the conclusions?

Reviewer #1: Yes

Reviewer #2: Partly

3. Has the statistical analysis been performed appropriately and rigorously?

Reviewer #1: Yes

Reviewer #2: Yes

4. Have the authors made all data underlying the findings in their manuscript fully available?

Reviewer #1: Yes

Reviewer #2: Yes

5. Is the manuscript presented in an intelligible fashion and written in standard English?

Reviewer #1: Yes

Reviewer #2: Yes

Reviewer #1: Dear author

I have no comment. All my comments were addressed and the paper can be accepted in the present form.

Congratulation

Reviewer #2: The paper is suitable rather than the first version, but it needs some minor revision in the introduction section. I think the below papers can help you to improve this section and point out the innovation:

Yu, Z., Zhao, J., Markov, V., & Han, D. (2024). Effects of hydrogen addition on ignition characteristics and engine performance of ammonia-hydrogen blended fuel: A kinetic analysis. International Journal of Hydrogen Energy, 87, 722-735.

Yao, Y., & Chen, S. (2016). A novel and simple approach to the good process performance of methane recovery from lignocellulosic biomass alone. Biotechnology for biofuels, 9, 1-9.

Yao, Y., Luo, Y., Yang, Y., Sheng, H., Li, X., Li, T., ... & An, L. (2014). Water free anaerobic co-digestion of vegetable processing waste with cattle slurry for methane production at high total solid content. Energy, 74, 309-313.

Yang, Z. Q., Hou, K. P., & Guo, T. T. (2011). Study on the effects of different water-cement ratios on the flow pattern properties of cement grouts. Applied Mechanics and Materials, 71, 1264-1267.

Gao, J., Wu, Y., & Shen, T. (2017). On-line statistical combustion phase optimization and control of SI gasoline engines. Applied Thermal Engineering, 112, 1396-1407.

I evaluated the reference section and found that ‘Zhang, Y., Xu, L., Wang, J., Pan, H., Dou, M., Teng, Y., ... & Wang, M. (2024). Bagasse-based porous flower-like MoS2/carbon composites for efficient microwave absorption. Carbon Letters, 1-16’ is in the reference section, but you wrote ‘Zhang et al, 2025’ in the text. Correct it to ‘Zhang et al, 2024’.

Use some more results in abstract section and improve it, so the readers can find total needed information.

T the paper can be accepted after the above correction.

**Do you want your identity to be public for this peer review?** For information about this choice, including consent withdrawal, please see our Privacy Policy

Reviewer #1: No

Reviewer #2: No

---

## [Author Response · Author response to Decision Letter 2]

21 Apr 2025

Response to Reviewers

Dear Editors and Reviewers,

We thank the Academic Editor and both reviewers for their valuable comments and positive evaluation of our revised manuscript. We are grateful for their time and effort, and we have addressed all the comments carefully as follows:

Reviewer #1:

Comment:

All my comments were addressed and the paper can be accepted in the present form. Congratulations.

Response:

We appreciate your positive feedback and are grateful for your acceptance of our revisions.

Reviewer #2:

Comment 1:

The introduction section needs minor improvement. The following references are suggested to strengthen the innovation aspects of the paper:

• Yu, Z., Zhao, J., Markov, V., & Han, D. (2024). Effects of hydrogen addition on ignition characteristics and engine performance of ammonia-hydrogen blended fuel: A kinetic analysis. International Journal of Hydrogen Energy, 87, 722-735.

• Yao, Y., & Chen, S. (2016). A novel and simple approach to the good process performance of methane recovery from lignocellulosic biomass alone. Biotechnology for biofuels, 9, 1-9.

• Yao, Y., Luo, Y., Yang, Y., Sheng, H., Li, X., Li, T., ... & An, L. (2014). Water free anaerobic co-digestion of vegetable processing waste with cattle slurry for methane production at high total solid content. Energy, 74, 309-313.

• Yang, Z. Q., Hou, K. P., & Guo, T. T. (2011). Study on the effects of different water-cement ratios on the flow pattern properties of cement grouts. Applied Mechanics and Materials, 71, 1264-1267.

• Gao, J., Wu, Y., & Shen, T. (2017). On-line statistical combustion phase optimization and control of SI gasoline engines. Applied Thermal Engineering, 112, 1396-1407.

Response:

Thank you for your valuable suggestion. In the revised version of the paper, we have carefully reviewed and incorporated the recommended literature to strengthen the introduction section. These additions have helped us better position our work within the existing research landscape and highlight the novel contributions of our study.

We believe the revised manuscript has significantly improved in clarity and scientific rigor, thanks to your valuable suggestions.

Comment 2:

In the text, the citation “Zhang et al., 2025” should be corrected to “Zhang et al., 2024.”

Response:

Thank you for spotting this mistake. We have corrected the citation in the main text to “Zhang et al., 2024.”

Comment 3:

Use more results in the abstract section and improve it so that readers can get the needed information.

Response:

We have revised the abstract by including more quantitative results and key findings from our analysis to provide a clearer and more informative summary for readers.

---

## [Editor Report · Decision Letter 2]

24 Apr 2025

Comparative LCA of diesel and grid electricity for agricultural irrigation in Iran: why diesel outperforms in fossil-reliant grids

PONE-D-25-10261R2

Dear Dr. Namdari,

We’re pleased to inform you that your manuscript has been judged scientifically suitable for publication and will be formally accepted for publication once it meets all outstanding technical requirements.

Kind regards,

Morteza Taki, Ph.D

Academic Editor

PLOS ONE

Additional Editor Comments (optional):

Dear Dr Namdari

I evaluated the paper. I think all the comments were addressed and the paper can be accepted in the present form.

---

## [Editor Report · Acceptance letter]

PONE-D-25-10261R2

PLOS ONE

Dear Dr. Namdari,

I'm pleased to inform you that your manuscript has been deemed suitable for publication in PLOS ONE. Congratulations! Your manuscript is now being handed over to our production team.

Kind regards,

on behalf of

Dr. Morteza Taki

Academic Editor

PLOS ONE